

# Distorted body schema after mastectomy with immediate breast reconstruction: a 4-month follow up study

Asall Kim[1,2], Eun Joo Yang[3], Myungki Ji[1], Jaewon Beom[1] and Chunghwi Yi[4]

[1] Department of Rehabilitation Medicine, Seoul National University Bundang Hospital, Seongnam, South Korea
[2] Department of Physical Therapy, The Graduate school, Yonsei University, Wonju, South Korea
[3] Department of Rehabilitation Medicine, Daelim Catholic Hospital, Seoul, South Korea
[4] Department of Physical Therapy, College of Software and Digital Healthcare Convergence, Yonsei University, Wonju, South Korea

Corresponding author
Chunghwi Yi, pteagle@yonsei.ac.kr

## ABSTRACT

**Background:** After breast cancer, some patients report residual pain-related upper limb disability without physical impairment. Although pain and altered proprioception are known to affect the working body schema (WBS), there is little available evidence investigating the WBS of breast cancer survivors (BrCS). WBS—body representations in the brain—affect the "neuromatrix" that modulates pain sensitivity and the threshold for threatening stimuli. The aim of this study was to investigate whether WBS was disrupted after mastectomy with immediate breast reconstruction (IBR) for breast cancer and whether pain and proprioceptive changes affected WBS.

**Methods:** Thirty-five BrCS participated in the 4-month follow-up study. They were observed at 1 and 4 months postoperatively. The main outcome measures were the left right judgement test (LRJT) results, absolute angle error, pectoralis minor length index (PMI), pain, and Quick-Disabilities of the Arm, Shoulder, and Hand (Q-DASH) score. They were measured at each observation, and parametric tests were performed to identify the nature of WBS.

**Results:** Both the reaction time and accuracy of the hand LRJT were poorer than those of the foot and back LRJT ($p < 0.001$). The hand LRJT reaction time and accuracy were unchanged over the total follow-up period ($p = 0.77$ and $p = 0.47$, respectively). There was a weak correlation between the LRJT reaction time and PMI ($r = -0.26$, $p = 0.07$), pain severity ($r = 0.37$, $p = 0.02$), and Q-DASH score ($r = 0.37$, $p = 0.02$). There was also a weak correlation between LRJT accuracy and Q-DASH score ($r = -0.31$, $p = 0.04$). The LRJT accuracy of BrCS who underwent surgery on their dominant side was higher than that of BrCS who underwent surgery on their non-dominant side ($p = 0.002$). Regression analysis found a weak but significant relationship between the early hand LRJT results and late pain severity (adjusted $R^2 = 0.179$, $p = 0.007$). A similar relationship was found between early hand LRJT results and Q-DASH score (adjusted $R^2 = 0.099$, $p = 0.039$).

**Conclusion:** To the best of our knowledge, this is the first study providing the nature of WBS after mastectomy with IBR. In this population, it is necessary to postoperatively preserve WBS integrity for pain and upper limb disability.

## INTRODUCTION

The increasing prevalence of breast cancer in young women has led to increased interest regarding upper limb function after surgery and treatment (*Kummerow et al., 2015*). Limitation of range of motion (LOM), decreased upper limb muscle strength (*Harrington et al., 2013*), shortened pectoralis muscles (*Lee et al., 2019*; *Yang et al., 2010*), and altered proprioception (*Zabit & Iyigun, 2019*) are commonly observed after breast cancer treatment. These factors affect each other and are related to upper limb disability (*Harrington et al., 2013*; *Lee et al., 2019*; *Yang et al., 2010*). Therefore, current studies recommend early free range of motion (ROM) exercises (*de Almeida Rizzi et al., 2020*) or immediate breast reconstruction (IBR) (*Myung et al., 2018*) for physical function recovery. However, many breast cancer survivors (BrCS) still complain of upper limb dysfunction without physical dysfunction (*Siqueira et al., 2021*). A recent study reported that pain components were associated with upper limb dysfunction (*Siqueira et al., 2021*).

There are several reasons for the occurrence of pain in this population; however, the reason for sustained pain in this population has rarely been discussed. In pain science, disrupted working body schema (WBS) is known to delay pain and disability resolution (*Moseley & Flor, 2012*). The WBS is stored in the sensory and motor cortices, and is able to recognize the size and orientation of body parts to execute movements precisely (*Holmes & Spence, 2004*). Since sensory input from the cortical representation of S1 affects the integrity of WBS, WBS disruption would alter the movement execution (*Bray & Moseley, 2011*; *Moseley & Flor, 2012*). The left right judgement test (LRJT) is the preferred tool for the evaluation of WBS disruption. The test assesses the reaction time and accuracy of discrimination when the participant is asked to decide the side of displayed body part images. The reaction time represents the processing time in the motor cortex (*Hudson et al., 2006*), whereas the accuracy represents the cortical proprioceptive representation (*Moseley & Flor, 2012*). Hand and foot LRJTs were performed to evaluate upper and lower limb body representation; currently, various body part discrimination tests—such as shoulder, knee, and movement direction of the neck and back—are available (*Breckenridge et al., 2019*). The assessment ability of LRJT depends on the affected body part; for example, the hand LRJT was not disrupted in participants with neck pain (*Wallwork et al., 2020*).

Disrupted WBS has been reported in those with neurologic impairment (*Conson et al., 2010*; *Fiori et al., 2014*), limb loss (*Nico et al., 2004*), proprioception alterations (*Meugnot & Toussaint, 2015*; *Silva et al., 2011*), and chronic pain conditions (*Breckenridge et al., 2019*). Although these factors are commonly observed in BrCS, only one study (*Boyd, Smoot & Nee, 2022*) has reported WBS disruption in this population. In this study, the LRJT results of the hand, shoulder, and chest were poorer than those of control groups and the chest LRJT was affected by various factors such as chemotherapy history, reconstructive surgery,

and pain-related components (*Boyd, Smoot & Nee, 2022*). Considering that the hand and shoulder LRJT ability represent the upper limb WBS (*Breckenridge et al., 2019*; *Breckenridge et al., 2020*), there was little evidence reporting the assessment ability of upper limb LRJT and related factors in this population (*Boyd, Smoot & Nee, 2022*). Furthermore, most of the study was conducted on chronic pain participants (*Barbosa et al., 2021*; *Bray & Moseley, 2011*; *Breckenridge et al., 2019*; *Breckenridge et al., 2020*; *Ismail et al., 2019*; *Magni, McNair & Rice, 2018*; *Pelletier et al., 2018a*; *Pelletier, Higgins & Bourbonnais, 2018b*; *Schmid & Coppieters, 2012*; *Wallwork et al., 2020*; *Wiebusch, Coombes & Silva, 2021a*); therefore, the effect of WBS disruption on the recovery of pain and disability was less understood.

BrCS commonly show reduced length of the pectoralis minor after surgery (*Lee et al., 2019*), which induces scapular protraction. Considering that altered scapular alignment affects shoulder kinematics and function (*Ha et al., 2016*; *Lee et al., 2015*; *Ludewig & Cook, 2000*; *Reinold, Escamilla & Wilk, 2009*), this positional and muscular change may evoke pain and decline in proprioceptive function (*Caldwell, Sahrmann & Van Dillen, 2007*; *Janwantanakul et al., 2002*; *Voight et al., 1996*). The presence of pain due to the surgery (*Bosompra et al., 2002*) could also contribute to posture changes owing to protective (*Lee et al., 2019*; *Stubblefield & Keole, 2014*), intended disuse (*Zocca et al., 2018*), as well as cortical reorganization (*Coslett et al., 2010*). Therefore, these abnormal sensory inputs could disrupt the upper limb WBS so that the movement execution would be adapted to the body representations (*Holmes & Spence, 2004*). Consequently, BrCS would perceive discomfort or disability, despite sufficient physical function. Therefore, we designed this follow-up study to verify this scenario with the perspective of WBS.

The first purpose of our study was to investigate the upper limb body schema disruption along the course of breast cancer treatment. We hypothesized that the hand LRJT results would be poorer than the foot (remote body region) and back LRJT results (different type of LRJT and movement directions) at early observations, but the differences would disappear at later observations. The second purpose of the study was to identify the predictive value of WBS for pain and disability. For this, we hypothesized that the early measured hand LRJT results (reaction time and/or accuracy) would be directionally associated with later pain severity and upper limb disability score. The third purpose of this study was to identify factors related to hand LRJT results. We hypothesized that there was a directional relationship between joint-reposition angle error, pectoralis minor length index (PMI), pain severity, disability level, and hand LRJT results. The extent of S1 representation depends on the use (*Gindrat et al., 2015*) and type of prosthesis (*Nico et al., 2004*). *Gindrat et al. (2015)* found continuous reshaping of sensory processing *via* repetitive hand movements. In this way, the use of the limb improves the cortical representation (*Gindrat et al., 2015*). Considering that the dominant arm is used more than the non-dominant arm after breast cancer surgery (*Fisher, Davies & Uhl, 2020*), the BrCS who underwent surgery on their dominant side (DS) might have better hand LRJT results than the BrCS who underwent surgery on their non-dominant side (NDS). *Nico et al. (2004)* found poor discrimination performance in amputees wearing prostheses compared to controls and amputees not wearing prostheses. In addition, they also reported

poorer performance in the amputees with aesthetic prostheses than those with myo-electric prostheses, which produce the actual movement. Based on their speculation that the aesthetic prostheses emphasized a mismatch between the motor command and the sensory feedback, BrCS who underwent direct-to implant or tissue expander insertion (DoT) would have poorer hand LRJT results than those who underwent transverse rectus abdominis myocutaneous (TRAM) flap reconstruction. This is because the implant or tissue expander provides poorer sensation recovery than does autologous breast reconstruction (*Hwang et al., 2022*).

## MATERIALS AND METHODS

### Study design and ethical approval

This study formed part of a cohort study observing BrCS after mastectomy with IBR in a clinical setting. This STROBE study was designed to observe BrCS at 1 and 4 months postoperatively. From August 2021 to March 2022, 67 participants were enrolled in the cohort study. Ethical approval was obtained from the Seoul National University Bundang Hospital Institutional Review Board (IRB No. B-2108-702-309). This study was also registered at the Clinical Research Information Service (Registration No. KCT0006501). All participants provided written informed consent per the guidelines of the Declaration of Helsinki.

### Participants

As the study design involved two-way repeated measure analysis of variance (3 (task) * 2 (time) RM ANOVA), the total sample size was calculated using the GLIMMPSE 3.0.0 online power and sample size calculation program (*Kreidler et al., 2013*). Using an alpha of 0.05 and power of 0.8, both Geisser–Greenhouse and Huynh–Feldt corrected tests recommended a sample size of 31. Considering a drop-out rate of 10%, 35 participants were required. Among the enrolled participants who provided informed consent, participants younger than 65 years without sustained pain in the leg and back were included in this study to study the effect of aging and pain on the back and foot LRJT results. We conducted additional LRJTs, physical assessments, and questionnaires on 35 BrCS at each visit. The second follow-up measurements ended in April 2022. The overall observation flow is shown in Fig. 1.

### Electronic medical record review

The electronic medical records were reviewed by A.K. to identify breast cancer surgery data (operation date and side, mastectomy type, lumpectomy type, and reconstruction type). Participants' height, weight, cancer treatment data, history of chemotherapy (yes/no), radiation therapy (yes/no), tamoxifen intake (yes/no), and presence of edema (yes/no) at each visit were also identified by A.K.

### LRJT

Region-specific WBS was measured using LRJT. The Recognise™ Hand, Back, and Foot applications (https://www.noigroup.com/product/recogniseapp/; Noigroup, Adelaide,

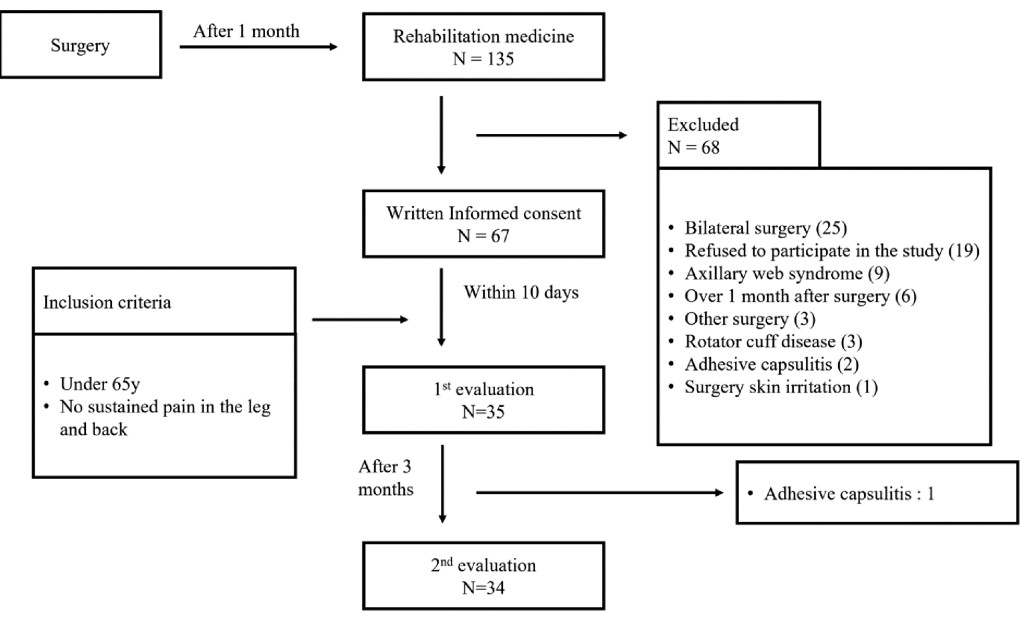

**Figure 1 Overall flowchart of the study.**

Australia) were administered using an iPad®. Random numbers were generated online (https://www.randomizer.org/). The order of applications was organized according to the random number. The software was set in the "vanilla" mode with 40 images and 5 s of display time for each image (*Wallwork et al., 2020*). The participant was asked to sit on a stool, and the iPad was placed on the table. To minimize orientation bias and hand dominance bias, the iPad was placed in front of the participant's midline. Next, the participant was asked to touch the "right" and "left" buttons with their right and left index fingers, respectively. To avoid copying the image, the participant was not allowed to move their body, but was instead requested to discriminate as quickly as possible. If the decision was not made within 5 s, the next image would appear automatically. Familiarization trials were provided for each application. For the familiarization trial, 20 images were displayed in the vanilla mode with a 5-s display time. After a brief rest (30 s), two test trials were conducted with rest time between trials (*Wallwork et al., 2020*). After two trials, a 1-min rest time was provided before performing the second-order application. Thus, a total of 240 images (80 * 3) were discriminated. The reaction time (s) and accuracy (%) of each side were automatically recorded by the application. In addition, results of the two sides were averaged and documented for analysis. The overall procedure was supervised by A.K.

## PMI

The PALpation Meter (PALM; Performance Attainment Associates, St Paul, MN, USA) was used to measure the distance between the coracoid process and the fourth intercostal space (*Harrington, Hoffman & Katsavelis, 2020*). This was measured three times by A. K., and the data were averaged. As the pectoralis minor length differs by height, the length was divided by the height for normalization.

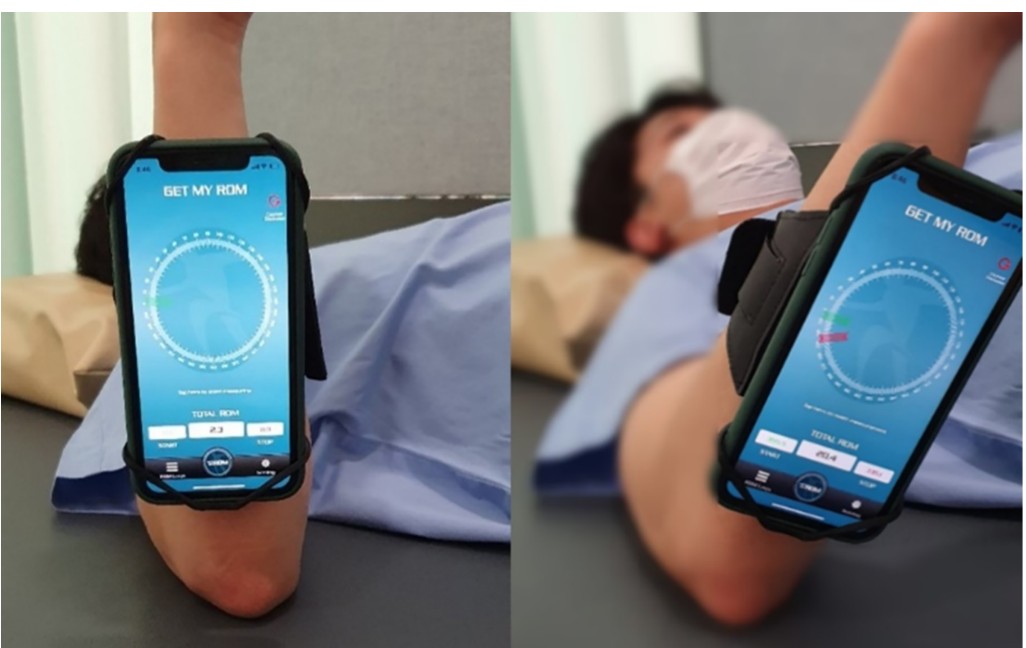

**Figure 2** Measurement of the joint reposition angle error.

## Joint-reposition angle error

The GetMyROM version 1.0 (https://apps.apple.com/kr/app/getmyrom/id438534405; Interactive Medical productions, Hampton, NH, USA) was administered using an iPhone 11® to observe real-time ROM. This mobile application is a reliable and validated goniometer to measure shoulder ROM (*Mejia-Hernandez et al., 2018*). In a previous study (*Ager et al., 2017*), a passive/active protocol of internal rotation in the supine position was performed. In the current study, the target angle was set at 10° to avoid the sense of tightness.

The test position is shown in Fig. 2. The arm band was fastened to hold the mobile phone, and then the arm was moved to 0° (start position). When the arm was in the start position, the screen was touched to record the start angle. For the test, the operated arm was moved to the target angle (10°), and the position was maintained for the participant to memorize. After 5 s of memorization, the arm was returned to the start position and the participant was asked to actively move to the target angle. When the participant felt they had reached the target angle, they were asked to stop and say "here." When the screen was touched to record the stop angle, the total ROM was automatically calculated. Three trials were conducted, and the absolute error angle (difference between 10° and the total ROM) was documented. The average absolute angle error was used for the analysis.

## Limitation of shoulder ROM

The participant's shoulder ROM was evaluated *via* the same application above. The active flexion and abduction angles were tested in the sitting position, whereas the active external rotation was tested in the supine position. According to the normative data, we regarded

150° and 60° as the cut-off values for normal elevation and external rotation, respectively (*Gill et al., 2020*). Thus, the average angle between flexion and abduction of <150° was regarded as elevation limitation. Likewise, external rotation of <60° was regarded as external rotation limitation. As pectoral tightness was the major complaint after breast cancer surgery, we excluded internal rotation LOM in this study.

## Questionnaires

The questionnaires for the dominant hand, pain characteristics, and disability were provided by MJ. Any missing item was requested to be filled out. The dominant hand was defined as the writing hand (*Shiri et al., 2007*); based on this, DS surgery (yes/no) was classified.

For pain evaluation, a simple questionnaire was provided: (1) present pain existence (yes/no), (2) visual analogue scale (VAS) for severe pain intensity lasting 1 month. The question regarding present pain existence was to identify whether they felt pain in the previous 1 week. VAS is an 11-point Likert scale (0–10, 10 indicates extreme pain such as pain during delivery) instrument to subjectively evaluate pain severity.

For disability evaluation, the Quick-Disabilities of the Arm, Shoulder, and Hand score (Q-DASH) was administered. The Q-DASH (Korean version (*Lee et al., 2008*)) is an easy and reliable tool that validates subjective assessment to assess upper limb disability in BrCS (*LeBlanc et al., 2014*). The Q-DASH score was normalized according to instruction (*Kennedy, 2011*). Although there is no definitive cut-off value to distinguish normal ability from upper limb disability, we defined 16 as the cut-off value—based on normative data (*Aasheim & Finsen, 2014*) of the Q-DASH score among 30- to 60-year-old women.

At the second visit, one exercise adherence grading questionnaire was added. The exercise adherence questionnaire consisted of three questions: (1) I practiced the exercise as instructed for the last 3 months, (2) I practiced both exercises, and (3) I followed the instructed exercise frequency. The response was graded on a 5-point Likert scale (0–4; never, rarely, sometimes, often, always). The normalized exercise adherence was calculated as the average of three responses multiplied by 25.

## Exercise education

After physical assessments, instructions for two exercises to improve pectoral muscle tightness and scapular stability were provided (M.J.). Detailed instructions are provided in the supplemental information. Additional exercises such as yoga, pilates, and general stretching were permitted.

## Statistical analysis

All statistical analyses were performed using SPSS 26.0 (SPSS, Chicago, IL, USA). Parametric tests were performed as all statistics met normality criteria (Shapiro–Wilk test and visual inspection of Q-Q plot). A correlation test was performed to determine aging effect and time-accuracy trade-off (*i.e.*, slower but correct response), which would indicate incorrect performance of LRJT. In all analyses, the alpha level was set at 0.05. Per-protocol analysis was performed. Two RM ANOVA was performed to determine within effects and
**Table 1 Participant age and cancer-related information.**

| Category | Frequencies |
|---|---|
| Age (years), mean (SD) | 45.23 (1.174) |
| Tumor stage (is/1/2/3) | 3/17/13/2 |
| Node stage (0/1/2/3) | 25/6/2/2 |
| Type of mastectomy | |
| Nipple sparing/Skin sparing/Total | 26/6/3 |
| Type of lymph node dissection | |
| None/SLNB/ALND/Both | 1/26/4/4 |
| Type of reconstruction | |
| TRAM/DoT | 17/18 |
| Surgery side (Right/Left) | 18/17 |
| Surgery on dominant side (yes/no) | 20/15 |

Note:
Results are expressed as frequencies unless otherwise specified. SD, standard deviation; is, carcinoma *in situ*; SLNB, sentinel lymph node biopsy; ALND, axillary lymph node dissection; TRAM, transverse rectus abdominis myocutaneous flap; DoT, direct-to-implant or tissue expander insertion.

an interaction effect for reaction time and accuracy. Within factors were time (2) and task (3). The LRJT results were pooled to conduct Bonferroni adjustment and Tukey's honestly significant difference (HSD) test for *post-hoc* analysis. Two linear regressions were performed to explain post-pain severity and post-Q-DASH score with early hand LRJT reaction time and accuracy. Given the recommendation of at least 10 samples per variable (*Kotrlik & Higgins, 2001*), 34 samples were enough to conduct analyses. The linear regression was performed with the stepwise method.

Pearson's correlation tests were performed to examine correlations between variables (absolute angle error, PMI, pain intensity, and Q-DASH score) and hand LRJT results at each visit. In addition, intraclass correlation coefficient (ICC) estimates for absolute angle error measurement and their 95% confidence intervals were calculated based on a single measurement, absolute-agreement, and a two-way mixed-effects model. For the two-way mixed-effects model, two-way RM ANOVA was performed to determine a main effect for reconstruction type and surgery side, and an interaction effect between the two variables.

# RESULTS

## Participants

Among 35 participants, 34 BrCS participated in the two outcome measurements. One BrCS could not participate in the second visit because of newly diagnosed adhesive capsulitis. Participant age and cancer-related information are shown in Table 1.

Over the follow-up period, physical variables such as LOM, joint position sense angle error, PMI, and Q-DASH score were significantly improved. However, the LRJT results and pain index remained unchanged. Table 2 summarizes all outcomes within the cohort.

## Aging effect and accuracy-time trade-off

The one-tailed Pearson's correlation test found no to weak correlation coefficients (r = −0.13–0.09) between age and hand LRJT results over time, and no correlation

**Table 2  Mean and standard deviation of the assessments.**

| Category | First visit | Second visit | p value |
|---|---|---|---|
| Sample size | 35 | 34 | N/A |
| Height (cm) | 161.65 (4.63) | 162.18 (4.63) | 0.00[a,**] |
| Weight (kg) | 58.13 (6.65) | 58.13 (6.21) | 0.93[a] |
| Postoperative day | 39.31 (7.66) | 119.50 (16.86) | 0.00[a,**] |
| History of chemotherapy (yes/no) | 18/17 | 20/14 | 0.69[b] |
| History of radiation therapy (yes/no) | 6/29 | 11/23 | 0.13[b] |
| History of tamoxifen intake (yes/no) | 11/24 | 17/17 | 0.15[b] |
| Presence of edematous arm (yes/no) | 5/30 | 6/28 | 1.00[b] |
| ROM limitation of elevation (yes/no) | 22/13 | 9/25 | 0.00[b,**] |
| ROM limitation of external rotation (yes/no) | 5/30 | 5/29 | 1.00[b] |
| LRJT reaction time (s) | | | |
| Hand | 1.92 (0.40) | 1.95 (0.45) | 0.77[a] |
| Back | 1.61 (0.38) | 1.58 (0.35) | 0.27[a] |
| Foot | 1.45 (0.36) | 1.41 (0.33) | 0.15[a] |
| LRJT accuracy (%) | | | |
| Hand | 78.71 (8.99) | 79.82 (7.19) | 0.47[a] |
| Back | 87.57 (8.59) | 88.68 (6.86) | 0.27[a] |
| Foot | 89.50 (7.32) | 92.32 (5.02) | 0.02[a,*] |
| Joint-reposition angle error (°) | 3.37 (2.18) | 2.06 (1.27) | 0.00[a,**] |
| Pectoralis minor length index | 9.81 (0.38) | 10.28 (0.29) | 0.00[a,**] |
| Present pain (yes/no) | 27/8 | 20/14 | 0.15[b] |
| VAS-severe pain (0–10, 0 means no pain) | 4.31 (2.54) | 4.18 (2.72) | 0.96[a] |
| Quick DASH score (0–100, 0 means no disability) | 28.77 (15.70) | 22.53 (16.35) | 0.02[a,*] |
| Upper limb disability (yes/no) | 27/8 | 18/16 | 0.04[b] |
| Exercise adherence score (0–100, 0 means no exercise adherence) | | 58.09 (20.57) | N/A |

**Notes:**
[a] Paired t-test (two-tailed).
[b] McNemar test.
* p-value < 0.05.
** p-value < 0.01.
Results are expressed as frequencies and mean (SD). SD, standard deviation; ROM, range of motion; LRJT, left right judgement test; VAS, visual analogue scale; DASH, disabilities of the arm, shoulder, and hand Quick DASH scores over 16 are classified into the upper limb disability group.

coefficient (r = 0.01–0.10) between age and foot LRJT results over time. There were only weak to moderate correlation coefficients (r = −0.49–0.49, p = 0.00–0.03) between age and back LRJT results over time. Using Pearson's correlation test, we investigated whether a time-accuracy trade-off existed; there were negative correlation coefficients (r = −0.14 to −0.58) between time and accuracy in all LRJTs over time. Thus, it was justified to not consider age as a co-variate, and the LRJTs were performed appropriately.

## Purpose 1: to evaluate WBS distortion and its change over time

Two-way (time*task) RM ANOVA was performed to determine the main and interaction effects within factors for reaction time and accuracy. For the main effect (time) of reaction time and accuracy, sphericity was met as indicated by Mauchly's test (Mauchly's $W = 1.000$). For the main effect (task) of reaction time and accuracy, sphericity was met as indicated by Mauchly's test ($\chi^2(2) = 2.57$, $p = 0.28$ and $\chi^2(2) = 3.37$, $p = 0.19$, respectively). For the time*task interaction effect of reaction time and accuracy, sphericity was met as indicated by Mauchly's test ($\chi^2(2) = 0.09$, $p = 0.95$ and $\chi^2(2) = 3.69$, $p = 0.16$, respectively). Repeated measures ANOVA for reaction time reported a main effect for the task (F(2, 66) = 61.65, $p = 0.00$, partial eta square = 0.65). There was no main effect for time (F(1,33) = 0.81, $p = 0.38$, partial eta square = 0.02), and no interaction effect between the task and time (F(2,66) = 1.07, $p = 0.35$, partial eta square = 0.03). Repeated measures ANOVA for accuracy also found a main effect for the task (F(2,66) = 41.33, $p = 0.00$, partial eta square = 0.56); however, there was no main effect for time (F(1,33) = 4.04, $p = 0.05$, partial eta square = 0.11), and no interaction effect between the task and time (F(2,66) = 0.72, $p = 0.49$, partial eta square = 0.02). Both Bonferroni adjustment and Tukey's HSD tests for multiple comparisons found that the mean value of the hand LRJTs (both reaction time and accuracy) was only significantly different between that of the back and foot in both evaluations ($p = 0.00$). The overall results of the analyses are shown in Fig. 3.

## Purpose 2: to study the relationship between early hand LRJT results and late pain/disability

According to linear regressions, each reaction time and accuracy at the first visit could solely predict pain severity and Q-DASH score at the second visit, respectively. The variance inflation factor for each of the two models was 1.000. The partial correlations between the hand LRJT results (reaction time and accuracy) at the first visit and pain severity at the second visit were r = 0.45 ($p = 0.004$) and r = −0.21 ($p = 0.12$), respectively. The partial correlations between the hand LRJT results (reaction time and accuracy) at the first visit and the Q-DASH score at the second visit were r = 0.27 ($p = 0.06$) and r = −0.36 ($p = 0.02$), respectively. Although the significant models were reported, the explanation power was weak (*Chin, 1998*). Results of regression analyses for each dependent variable are described in Table 3.

## Purpose 3: to identify factors affecting hand LRJT results

At the first visit, there were no significantly correlated variables. The absolute error angle had no correlation with reaction time and accuracy (r = −0.06 and −0.01, $p = 0.36$ and 0.47, respectively). Additionally, PMI had a very weak negative correlation with reaction time and accuracy (r = −0.19 and −0.16, $p = 0.14$ and 0.18, respectively). Pain severity (VAS) had a very weak positive correlation with reaction time and accuracy (r = 0.18 and 0.11, $p = 0.15$ and 0.27, respectively). The Q-DASH score also had a very weak correlation with reaction time and accuracy (r = 0.14 and −0.13, $p = 0.22$ and 0.22, respectively).

At the second visit, there were weak and significantly correlated variables. The absolute error angle showed no correlation with reaction time and accuracy (r = −0.04 and 0.08,

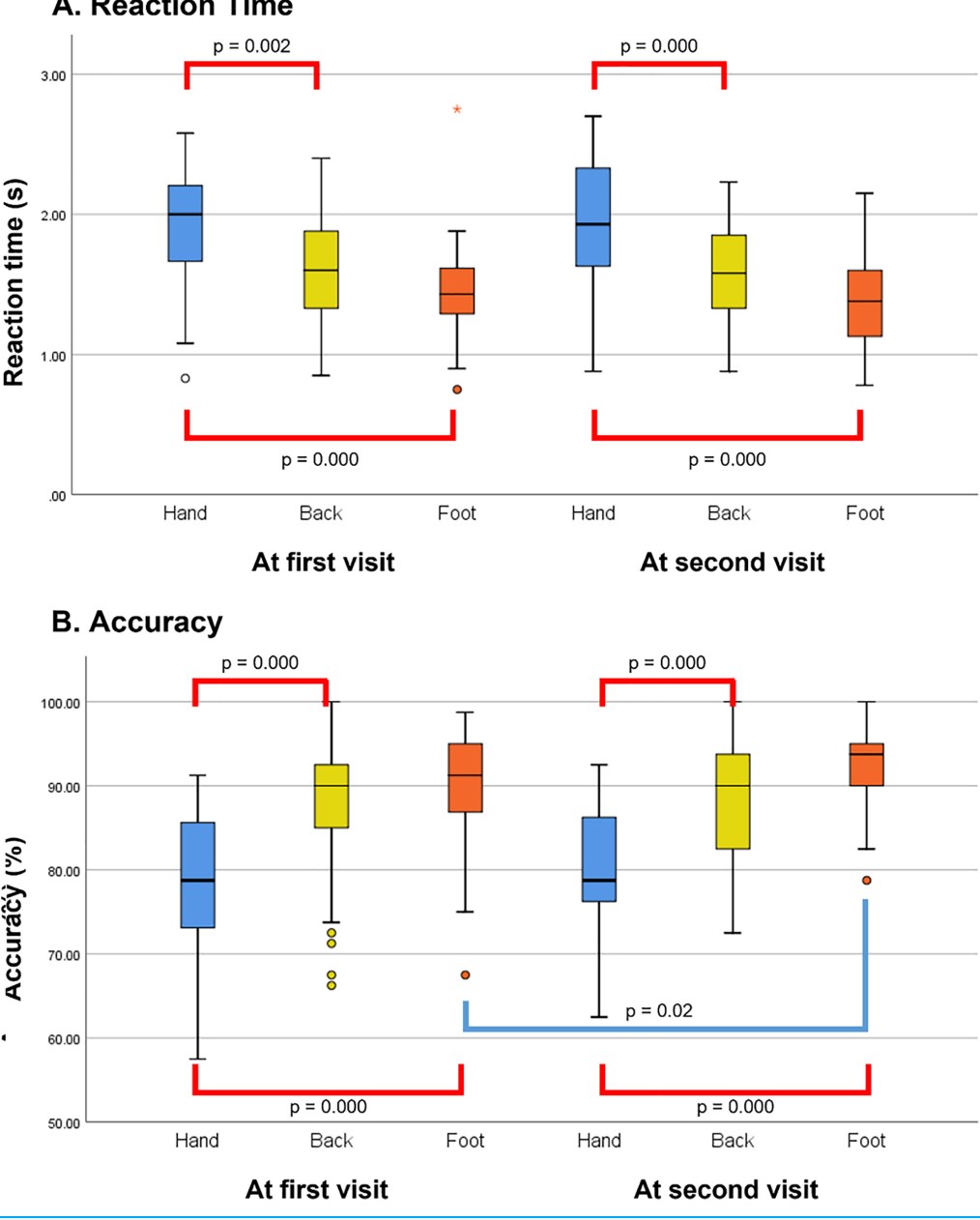

**Figure 3 Group differences in the left/right judgement test over the follow-up period.** (A) Group differences in the reaction time. (B) Group differences in the accuracy. The red line indicates the significant *post-hoc* Bonferroni analysis and Tukey's test findings for this comparison, performed following an overall main effect of task. The *p*-value of both *post-hoc* analyses were identical. The blue line indicates the significant paired t-test findings.

*p* = 0.41 and 0.32, respectively). The PMI had a weak negative correlation with the reaction time (r = −0.26, *p* = 0.07); however, there was no correlation with accuracy (r = −0.07, *p* = 0.36). Pain severity (VAS) correlated weakly, yet significantly, with reaction time (r = 0.37, *p* = 0.02); however, the correlation of VAS with accuracy was weak and insignificant (r = −0.19, *p* = 0.14). The Q-DASH score had a weak and significant

**Table 3 Results of the regression analyses.**

| Variables | Unstandardized coefficient B | Standard error | Standardized coefficient B | t | Significance |
|---|---|---|---|---|---|
| (A) Regression model for pain severity | | | | | |
| R = 0.452 | $R^2$ = 0.204 | Adjusted $R^2$ = 0.179 | | F(1,32) = 8.212 | 0.007** |
| Constant | −1.961 | 2.183 | | −0.898 | 0.376 |
| RT[a] | 3.165 | 1.104 | 0.452 | 2.866 | 0.007** |
| (B) Regression model for the Q-DASH score | | | | | |
| R = 0.356 | $R^2$ = 0.126 | Adjusted $R^2$ = 0.099 | | F(1,32) = 4.632 | 0.039* |
| Constant | 72.658 | 23.445 | | 3.099 | 0.004** |
| ACC[b] | −0.637 | 0.296 | −0.356 | −2.152 | 0.039* |

Notes:

[a] One-second increase in reaction time at 1 month postoperatively, associated with a 3.165-point higher severe pain intensity.

[b] One percent point increase in accuracy at 1 month postoperatively, associated with a 0.637-point lower Q-DASH score.

* $p$-value < 0.05.

** $p$-value < 0.01.

RT, reaction time at 1 month postoperatively; ACC, accuracy at 1 month postoperatively; Q-DASH, quick disabilities of the arm, shoulder, and hand.

correlation with reaction time and accuracy (r = 0.37 and −0.31, $p$ = 0.02 and 0.04, respectively). A poor-to-moderate degree (*Koo & Li, 2016*) of reliability was found between the absolute angle error measurements over the follow-up period. The single measure ICC (3,1) was 0.60 with a 95% confidence interval (CI) of 0.42–0.75 (F(34,68) = 5.66, $p$ = 0.00) at the first evaluation and 0.42 with a 95% CI of 0.21–0.62 (F(33,66) = 3.25, $p$ = 0.00) at the second evaluation.

Two-way RM ANOVA found no main effect of both reconstruction type and surgery side on reaction time (F(1,30) = 2.48, $p$ = 0.13 and F(1,30) = 1.35, $p$ = 0.26, respectively). Two-way RM ANOVA also found no interaction effect of reaction time between reconstruction type and surgery side (F(1,30) = 2.68, $p$ = 0.11). Although two-way RM ANOVA reported no main effect of reconstruction type on accuracy (F(1,30) = 0.01, $p$ = 0.91), there was a main effect of surgery side on accuracy (F(1,30) = 11.06, $p$ = 0.00). However, there was no interaction effect for accuracy between the two variables (F(1,30) = 0.02, $p$ = 0.90). The estimated marginal mean (EM mean) of the accuracy of DS (M = 82.42, 95% CI [79.50–85.34]) was significantly higher than that of NDS (M = 75.25, 95% CI [71.96–78.54]) ($p$ = 0.002). In addition, there was no interaction effect between time and surgery side (F(1,30) = 1.77, $p$ = 0.19). The EM mean of the accuracy of the DS at the first visit was 82.78 (95% CI [78.91–86.65]), whereas that of the DS at the second visit was 82.06 (95% CI [78.78–85.34]). The EM mean of accuracy of the NDS at the first visit was 73.57 (95% CI [69.21–77.93]), whereas that of the NDS at the second visit was 76.93 (95% CI [73.23–80.62]).

# DISCUSSION

This study aimed to investigate WBS after mastectomy with IBR using LRJT and to identify factors associated with WBS. The participants were in their forties. Considering the highest incidence rate in individuals aged 40–49 years in South Korea (*Kang et al.,*

2020), the sample could be representative of the population. During the follow-up, participants' physical variables such as arm elevation limitation, reposition angle error, PMI, and Q-DASH score were improved, whereas the pain severity and hand LRJT results remained unchanged. Interestingly, only a few BrCS showed the external rotation LOM; this may indicate that cancer treatment, including chemotherapy, radiation therapy, and tamoxifen intake, did not evoke capsular restriction. In addition, the instructed exercises likely contributed to improve physical variables, but not the WBS

## Primary findings

The first purpose of the study was to define differences between the results of LRJTs (hand, foot, and back). Based on our results, our hypotheses were partially proved. Two-way RM ANOVA and *post-hoc* analysis revealed a slower reaction time and poorer accuracy of the hand LRJT than that of the other two tasks. As we did not include a control group in this study, these additional comparisons should be performed referring to previous studies. One study compared the hand LRJT results of BrCS to that of healthy controls; this study reported reaction times and accuracies of 2.842 s and 81.46% for healthy controls and 3.229 s and 76.28% for BrCS, respectively (*Boyd, Smoot & Nee, 2022*). Our study results showed faster reaction times and similar accuracy in BrCS compared with the findings of this previous study (*Boyd, Smoot & Nee, 2022*). These differences may be owing to the older age of the control group in the previous study, longer response times provided (8,000 ms), and the possibility of the image being copied by the participant owing to the test environment (*Boyd, Smoot & Nee, 2022*). However, since previous studies have reported a 2 s reaction time and 90% accuracy within the no pain group in hand LRJT (*Breckenridge et al., 2019*; *Wallwork et al., 2020*), it is reasonable to report that BrCS showed poor proprioceptive representations during follow-up. In our present study, the processing time of the BrCS was not severely delayed (1.9 s reaction time); however, they had poor accuracy (80%) during the hand LRJT. In addition, the discrimination ability did not improve over the follow-up period. According to a previous study (*Harms et al., 2020*) comparing the effects of standard care and brain-targeted intervention for knee osteoarthritis, accuracy improved in the standard care group, whereas in the brain-targeted intervention group, accuracy was maintained (*Harms et al., 2020*). The author stated that standard care—including strengthening and mobilization—might require participants to pay close attention to their knee, and regular exercise might improve proprioceptive input, subsequently increasing the accuracy (*Harms et al., 2020*). However, performing pectoral stretching and strengthening the scapular stabilizer would not be sufficient to improve discrimination ability. Considering the recommendation to restore WBS (through targeted intervention) for limb and face conditions, but not for back and neck conditions (*Breckenridge et al., 2019*), this population would require exercise and brain-targeted interventions.

The second aim of the study was to find the predictive value of the hand LRJT for future pain and upper limb disability. Based on our results, the hypothesis was proven correct. Each hand LRJT reaction time and accuracy at the first visit significantly predicted pain severity and upper limb disability (Q-DASH score) at the second visit, respectively.

Disrupted WBS is reportedly associated with the fear of movement, catastrophizing (*Araya-Quintanilla et al., 2020*), and declined cognition (*Pelletier et al., 2018a*). Since WBS is the cortical representation, complex interactions between the physical body and neuro-matrix may have modulated the subjective symptoms. In other words, the patients' subjective evaluation of pain or disability could be devaluated because of the interaction. As people usually make use of their dominant arm after any surgery, the sense of movement success could reduce the feeling of pain or disability. Owing to the low explanation power, our study did not show WBS to be a powerful predictor of future pain and disability. However, it is worth considering WBS at the early stages to facilitate improvement of pain and disability in rehabilitation intervention after mastectomy with IBR. In addition, there were significant correlations between LRJT accuracy and Q-DASH score at the second visit. Over the follow-up period, nine participants reported limitation of arm elevation, whereas 18 reported upper limb disability. Based on these results, the LRJT should be evaluated from the first postoperative month to provide preventive or curative rehabilitation.

## Secondary findings

For the last purpose of this study, we investigated factors affecting the hand LRJT results over the follow-up period. We hypothesized that various postoperative factors might affect the integrity of WBS. Based on the correlation coefficient, our hypothesis that WBS would be directly associated with pain severity and disability level was partially supported. Although a different LRJT was used, *Boyd, Smoot & Nee (2022)* reported a regression model predicting chest LRJT results with various components. In the study, DASH score was one of the variables predicting accuracy, whereas the pain severity—using brief pain inventory—was one of the variables predicting reaction time (*Boyd, Smoot & Nee, 2022*). *Breckenridge et al. (2020)* also reported a significant regression model predicting shoulder LRJT results with current pain and disability level using the Shoulder Pain and Disability Index (SPADI) (*Breckenridge et al., 2020*). Although heterogeneity of pain duration was present, reaction time and accuracy were increased and decreased, respectively, when the current pain intensity increased (*Breckenridge et al., 2020*). Only the accuracy decreased when the SPADI score increased, which indicates severe disability level (*Breckenridge et al., 2020*). In contrast, another study (*Barbosa et al., 2021*) reported no correlation between pain intensity and shoulder LRJT results within chronic shoulder pain conditions (*Barbosa et al., 2021*). However, there are studies that reported no correlation between LRJT results and pain intensity or disability level in populations with upper limb pain conditions such as lateral elbow tendinopathy (*Wiebusch, Coombes & Silva, 2021a*), unilateral carpal tunnel syndrome (*Schmid & Coppieters, 2012*), wrist/hand disorder (*Pelletier, Higgins & Bourbonnais, 2018b*), and hand osteoarthritis (*Magni, McNair & Rice, 2018*). Therefore, the results of these previous studies indicate that the correlation between pain intensity and upper limb disability depends on the condition and pain duration. Our present study results, which report a significant correlation between LRJT results and pain intensity or Q-DASH score in BrCS, were only significant at 4 months postoperatively, and not at 1 month postoperatively. However, we failed to support our hypothesis that the reposition

angle error, which represents proprioception and pectoralis minor length, would have an impact on the hand LRJT results. In addition, there were very weak to no correlations at the first visit, whereas the correlations at the second visit were weak and significant. The reason we did not find any correlations in the early observation may be because the most effective factors in this stage would be sensory deficits or psychological changes, rather than the variables of interest in this study. After surgery, patients commonly complain of numbness (*Bosompra et al., 2002*), concerns of movement (*Van der Gucht et al., 2020*), and fear of recurrence (*Koch et al., 2014*). Compared with the arm on the contralateral side, the operated arm is therefore commonly moved less. A previous study showed that activity in the non-dominant arm was significantly lower than that in the dominant arm postoperatively (*Fisher, Davies & Uhl, 2020*). Furthermore, we did not find a significant correlation between WBS and proprioceptive components such as reposition angle error and PMI at the second visit. We speculate that altered proprioceptive accuracy and scapular position might be representative of the altered proprioceptive cortical maps. This might be because the source of the proprioceptive input was not peripheral. Previous studies reported no disruption of WBS in participants with ligament deficits (*Ismail et al., 2019*) and lateral elbow tendinopathy, who showed altered joint position sense (*Wiebusch, Coombes & Silva, 2021b*). However, other studies have reported a slower reaction time after hand immobilization (*Meugnot, Agbangla & Toussaint, 2016*; *Meugnot & Toussaint, 2015*; *Toussaint, Meugnot & Bidet-Ildei, 2021*). Given the results of our present study, as well as those of previous studies, the cortical map alterations may be due to the disuse of or decreased activity level of the upper limb, not the accuracy of such. This assumption may be supported by the significant main effect of the surgery side. In this study, the BrCS who underwent surgery on their DS performed the hand LRJT more accurately than those who underwent surgery on their NDS. Even though the accuracy was improved in the NDS group, this improvement was not significant. Although the instructed exercise would increase the activity level, it might be insufficient in the NDS group. In contrast, accuracy was maintained in the DS group, which might show that they have already used their dominant arm to some extent, and the exercise did not change the activity level. Thus, it is possible to conclude that insufficient movement of the limb may affect WBS integrity.

In addition, *Nico et al. (2004)* and *Meugnot & Toussaint (2015)* reported that the effect of limb loss and 48-h hand immobilization on LRJT was larger in the dominant hand, as the dominant hand was more affected by the level of physical activity (*Meugnot & Toussaint, 2015*). *Nico et al. (2004)* also reported the effect of wearing prostheses in upper limb amputees; the amputees wearing prostheses performed LRJTs more poorly than did controls and those not wearing prostheses. In addition, there were differences in LRJT results between the different types of prostheses. Two amputees wearing myo-electric prostheses, which allow specific thumb and wrist movements through residual forearm muscle contractions, performed slightly better than did other amputees wearing aesthetic prostheses. Based on this previous study, we formulated our hypothesis predicting better LRJT results in the TRAM group than that in the DoT group. However, there was no difference between the two groups in our study. This might be because the upper limb usage was not dependent upon the breast reconstruction material. In conclusion, the

surgery side (DS or NDS), but not the reconstruction type (TRAM or DoT), should be considered when evaluating LRJT in BrCS.

To the best of our knowledge, this is the first cohort study to investigate WBS in BrCS who underwent mastectomy with IBR. Despite the very weak to weak correlations, we investigated the relationships between LRJT results and physical functions that affect upper limb disability. Our study findings indicate that it is worth investigating WBS in BrCS who underwent mastectomy with IBR, and targeted intervention for WBS may be effective in improving upper limb pain and disability. In addition, the homogeneity within the population, as well as the timing of the evaluation, strengthened the value of this study.

This study has various limitations that need to be considered for interpretation. First, we did not include a control group (healthy control or surgery-type control). Therefore, additional comparison with previous studies is necessary to confirm our results regarding disrupted hand WBS. Second, we modified the method to evaluate the absolute angle error. We analyzed the test-retest reliability to cover this limitation. However, there were poor-to-moderate ICC (3,1). Furthermore, we did not assess other sensory aspects such as the two-point discrimination test and upper limb activity level. Therefore, we could only assume the relationship between proprioception and WBS. Lastly, we only followed up for 4 months after surgery. It is unclear whether this result would be the same at 4–5 months or more postoperatively.

## CONCLUSIONS

We demonstrated the distorted WBS and its associated factors in BrCS who underwent mastectomy with IBR. The surgery side (DS or NDS), as well as increases in pain intensity and disability level, were shown to alter WBS integrity. This study indicates that mastectomy with IBR may be accompanied by maladaptive proprioceptive map changes; therefore, postoperative evaluation and targeted intervention of WBS may be useful in this population. This finding reporting a significant correlation between LRJT accuracy and Q-DASH score at later observation (when many BrCS still reported upper limb disability without arm elevation limitation) also provides evidence for upper limb disability without shoulder ROM limitation. Considering the limitations of our present study, future studies investigating the effect of WBS-targeted intervention on upper limb pain and disability in this population are necessary to confirm our results. In future studies, the very early postoperative effect of brain-targeted intervention on pain, ROM, and upper limb disability improvement should be investigated to recommend the specific intervention for the immobilized phase after surgery.

## ACKNOWLEDGEMENTS
We would like to thank Editage for English language editing.

### Funding
The authors received no funding for this work.

## Competing Interests

The authors declare that they have no competing interests.

## Author Contributions

- Asall Kim conceived and designed the experiments, performed the experiments, analyzed the data, prepared figures and/or tables, authored or reviewed drafts of the article, and approved the final draft.
- Eun Joo Yang conceived and designed the experiments, authored or reviewed drafts of the article, project administration, and approved the final draft.
- Myungki Ji conceived and designed the experiments, performed the experiments, authored or reviewed drafts of the article, and approved the final draft.
- Jaewon Beom analyzed the data, authored or reviewed drafts of the article, project administration, and approved the final draft.
- Chunghwi Yi conceived and designed the experiments, analyzed the data, authored or reviewed drafts of the article, project administration, and approved the final draft.

## Human Ethics

The following information was supplied relating to ethical approvals (*i.e.*, approving body and any reference numbers):

Ethical approval was obtained from the Seoul National University Bundang Hospital Institutional Review Board (IRB No. B-2108-702-309).

## Data Availability

The raw data is available in the Supplemental Files.

## Supplemental Information

Supplemental information for this article can be found online at http://dx.doi.org/10.7717/peerj.14157#supplemental-information.

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
