# Peer review of "Distorted body schema after mastectomy with immediate breast reconstruction: a 4-month follow up study"

_PeerJ, doi:10.7717/peerj.14157_

## Round 0.1 · original submission · Minor Revisions

Dear authors,

Please reply point by point to reviewer 2's comments.

·

Basic reporting

The study is based on interesting and current topics. Hypotesis is underpinned by an up-to-date and compelling background.

Experimental design

The methodology adopted to verify the three hypotheses is well organized. The results, even when they have not fully confirmed all three hypotheses, are descriptive and convincing. The manuscript can be accepted in its current form.

Validity of the findings

The results, even when they have not fully confirmed all three hypotheses, are descriptive and convincing. The manuscript can be accepted in its current form.

Reviewer 2 ·

Basic reporting

The study is interesting and complies with the standards of the journal. The article includes sufficient introduction and background to demonstrate the purpose. It is written correctly

Experimental design

Overall, the methodology is clearly explained. The tools used are validated and reliable.
This section is well written, but the purpose of the study is unclear. I suggest inserting the phrase "the purpose 1 (2,3) of the study was ...”, as in the results section.

Validity of the findings

no comment. The discussions are clear and to point.

Additional comments

SPECIFIC COMMENTS
Abstract
It is written correctly. Gives highlights from each section of the paper.

Introduction

This section is well written, but the purpose of the study is unclear. I suggest inserting the phrase "the purpose 1 (2,3) of the study was ...”, as in the results section.

Methods

Overall, the methodology is clearly explained. The tools used are validated and reliable.
The statistical techniques used are appropriate.
However, in table 2 the Paired t-test is reported but, in the text, the Two RM ANOVA, I would suggest standardizing.

The authors declare an observational study design. I would suggest inserting the appropriate Strobe checklist and flow
https://www.strobe-statement.org/checklists/


Results

The results are clear and appropriate
Discussion
The discussions are clear and to point. The limitations are described.
Conclusions
The authors' conclusions are justified. The take-home message is clear.

---

## Round 0.2 · accepted · Accept

The authors addressed the reviewers' comments correctly and the changes are adequate; therefore, the manuscript deserves to be published.